# Slip Estimation and Compensation Control of Omnidirectional Wheeled Automated Guided Vehicle

Pei-Jarn Chen , Szu-Yueh Yang, Yen-Pei Chen, Muslikhin Muslikhin  and Ming-Shyan Wang *

Department of Electrical Engineering, Southern Taiwan University of Science and Technology, 1, Nan-Tai St., Yung Kang District, Tainan City 710, Taiwan; cpj@stust.edu.tw (P.-J.C.); Da720201@stust.edu.tw (S.-Y.Y.); ma620209@stust.edu.tw (Y.-P.C.); muslikhin@uny.ac.id (M.M.)

* Correspondence: mswang@stust.edu.tw; Tel.: +886-6-2533131 (ext. 3328)

**Abstract:** To achieve Industry 4.0 solutions for the networking of mechatronic components in production plants, the use of Internet of Things (IoT) technology is the optimal way for goods transportation in the cyber-physical system (CPS). As a result, automated guided vehicles (AGVs) are networked to all other participants in the production system to accept and execute transport jobs. Accurately tracking the planned paths of AGVs is therefore essential. The omnidirectional mobile vehicle has shown its excellent characteristics in crowded environments and narrow aisle spaces. However, the slip problem of the omnidirectional mobile vehicle is more serious than that of the general wheeled mobile vehicle. This paper proposes a slip estimation and compensation control method for an omnidirectional Mecanum-wheeled automated guided vehicle (OMWAGV) and implements a control system. Based on the slip estimation and compensation control of the general wheeled mobile platform, a Microchip dsPIC30F6010A microcontroller-based system uses an MPU-9250 multi-axis accelerometer sensor to derive the longitudinal speed, transverse speed, and steering angle of the omnidirectional wheel platform. These data are then compared with those from the motor encoders. A linear regression with a recursive least squares (RLS) method is utilized to estimate real-time slip ratio variations of four driving wheels and conduct the corresponding compensation and control. As a result, the driving speeds of the four omnidirectional wheels are dynamically adjusted so that the OMWAGV can accurately follow the predetermined motion trajectory. The experimental results of diagonally moving and cross-walking motions without and with slip estimation and compensation control showed that, without calculating the errors occurred during travel, the distances between the original starting position to the stopping position are dramatically reduced from 1.52 m to 0.03 m and from 1.56 m to 0.03 m, respectively. The higher tracking accuracy of the proposed method verifies its effectiveness and validness.

**Keywords:** omnidirectional Mecanum-wheeled automated guided vehicle (OMWAGV); recursive least square (RLS); slip ratio

## 1. Introduction

In recent years, with the prevailing trend of Industry 4.0, factories have increasingly higher requirements for the efficiency of warehouse systems. It is necessary to use Internet of Things (IoT) technology for networking among the production components and transportation vehicles [1–3]. Traditional vehicles use differential wheels, which are less efficient for factory handling. For example, when entering the station for goods, vehicles need to turn to face the entry site and then move forward. In addition, if the path layout includes a right angle, the differential wheeled vehicles cannot achieve walking by translation, which will be programmed to turn 90 degrees and then execute the straight walk command. Omnidirectional mobile wheeled vehicles have improved performance in congested environments and narrow aisles, which are commonly found in factory workshops, warehouses, offices, hospitals, etc. In fact, the topics of experimental investigations of a highly maneuverable mobile omniwheel robot, integration of inertial sensor data

into control of the mobile platform and navigation control, and stability investigation of a mobile robot based on a hexacopter equipped with an integrated manipulator have attracted much attention in Europe and around the world. As a result, the omnidirectional Mecanum-wheeled automated guided vehicle (OMWAGV) walks more flexibly along a path, and its walking mode includes going straight, translation, and oblique moving. If it encounters a right-angle path, the OMWAGV only needs to translate into the station. However, OMWAGVs slip more severely than differential wheeled vehicles.

A traction control system for vehicles is designed to prevent their performance from being degraded due to vehicle tire locking and skidding. Therefore, in harsh environments, such as slippery, snowy, and icy ground, the performance and stability of a vehicle is significantly improved by the use of a traction control system. A parameter identification technique with the constraint of minimizing power consumption for a city bus equipped with a permanent magnet three-phase synchronous motor is proposed in [4] to maintain performance. Furthermore, slip between the tire and the road surface greatly reduces speed. The required traction force depends on the slip ratio, the normal force acting on the tire, and the friction coefficient between the tire and the ground [5,6]. The friction coefficient has a very close relationship with the slip ratio under the ground–tire contact. Because the friction coefficient of the tire against the ground is unknown and changes with time during driving, the slip estimation based on data obtained by the sensors is highly important. The tire angular acceleration can easily be measured by a sensor, but the vehicle speed can only be calculated from the slip ratio.

The Mecanum wheel has excellent omnidirectional mobility and has attracted a lot of attention in the industry [7]. For example, under the prevailing trend of Industry 4.0, AGVs with Mecanum wheels are widely used in automated warehouse logistics systems and mobile robotic arms [8]. It also has various home, hospital, nuclear power plant, and military applications [9]. In addition to the diversity of its applications, most studies are devoted to eliminating the uncertainty inherent in the Mecanum wheel, such as the low and non-fixed friction to the ground, the displacement under contact points, abnormal wheel diameter, vibration during movement, etc.

Three typical tire–road friction coefficient estimation methods, the slip slope, individual tire force estimation, and extended Kalman filter, are reviewed and compared and then a new cost-effective tire–road friction coefficient estimation method is presented in [10]. In [5], the authors investigate the tire–road adhesion stability by observing the force transmitting behavior between the wheel drive torque to the tire–road adhesion torque. A closed-loop reduced-order observer is designed to estimate the adhesion torque in order to determine the optimal operation point for wheel slip prevention. To solve the problem of skid braking and spin acceleration, a second-order sliding-mode traction controller and a sliding-mode observer to estimate the tire–road adhesion coefficient are presented in [6]. The authors in [11] propose a sliding-mode controller with a conditional integrator for wheel slip in electric vehicles. Three different observers utilizing engine torque, brake torque, and global positioning system (GPS) measurements are designed to estimate the slip ratios and longitudinal tire forces in [12]. A recursive least squares (RLS) method is then used to identify the friction coefficient. The authors of [13] propose a tire–road friction coefficient estimation algorithm that utilizes the lateral dynamics of the vehicle, which is a function of slip angle, friction coefficient, normal force, and cornering stiffness. A differential GPS and a gyroscope are used to identify the real-time tire–road friction coefficient and cornering stiffness parameters of the tire. Based on a linear vehicle model and sensor measurements, the RLS method with a forgetting factor is utilized to estimate the vehicle sideslip. In addition, by integrating sensor measurements and roll dynamics, a Kalman filter is designed to estimate the roll angle in [14]. This methodology is also applied to estimate the nanosized parameters of a single-input and single-output linear model in [15]. A proposed wheel slip control system consists of three parts to maximize the braking force and maintain vehicle stability in [16]. These are a braking monitor based on the extended Kalman filter to estimate the tire braking force, lateral tire force and brake disc-pad friction

coefficient, and a sliding-mode wheel slip controller, and an optimally designed target slip assignment method. Using a different method, [17] proposes a geometric approach to obtain the decoupling controllability between horizontal, vertical, and angular motions and their functional controllability of the DC-drives for the Robotino.

The movement of a four Mecanum-wheeled mobile robot is controlled by the inverse kinematic so as to convert the robot velocity components, robot angular velocity, angular velocity of each wheel, and turn direction [18]. A kinematic controller for a Mecanum-wheeled omnidirectional robot is designed based on the feedback linearization method in an FPGA [19]. A robust and adaptive sliding mode controller for trajectory tracking of a Mecanum-wheeled mobile robot with uncertainties is proposed in [9]. The authors of [8] propose an adaptive, nonsingular terminal sliding mode control by the output recurrent fuzzy wavelet neural networks for a group of networked heterogeneous Mecanum-wheeled omnidirectional robots with uncertainties. However, they only conducted simulations. A Mecanum-wheeled vehicle with a vision sensor system is proposed to smoothly correct the vehicle position by continuously entering sensor data into the position feedback loop in [20]. The authors identify the positional error sources of Mecanum-wheel-based omnidirectional mobile robots due to wheel slip, and then adjust the wheel parameters to reduce positional errors in [21]. The authors of [22] propose the use of a hierarchical linear quadratic regulator (LQR) to satisfy the global objective (vehicle motion) and local objective (driving force and slip control of each wheel) of a Mecanum-wheeled vehicle.

In this paper, the OMWAGV modeling is introduced in Section 2. In Section 3, the method of slip estimation and compensation is described. Experimental results are shown in Section 4. Finally, conclusions are given in Section 5.

## 2. Modeling of Omnidirectional Mecanum-Wheeled Automated Guided Vehicle

The configuration of the OMWAGV is presented in Figure 1 [18]. With the different directions and speeds of the four wheels, the movement posture of the OMWAGV can be changed to move in all directions. The vehicle speed $v$ is composed of the longitudinal speed $v_x$ and the translational axis speed $v_y$ as follows [18]:

$$v_x = v \cos \theta \tag{1}$$

$$v_y = v \cos \theta \tag{2}$$

where $\theta$ is the yaw angle of the vehicle, $\omega$ is the steering angular speed, $r$ is the radius of the wheel, $\omega_i (i = 1, 2, 3, 4)$ is the angular velocity of each wheel, $v_i = r\omega_i (i = 1, 2, 3, 4)$ is the moving speed of each wheel, and the size of the vehicle is determined by the distance between the center of the body and the axle represented by the vectors **a** ($\{a_i, i = 1, 2, 3, 4\}$) and **b** ($\{b_i, i = 1, 2, 3, 4\}$).

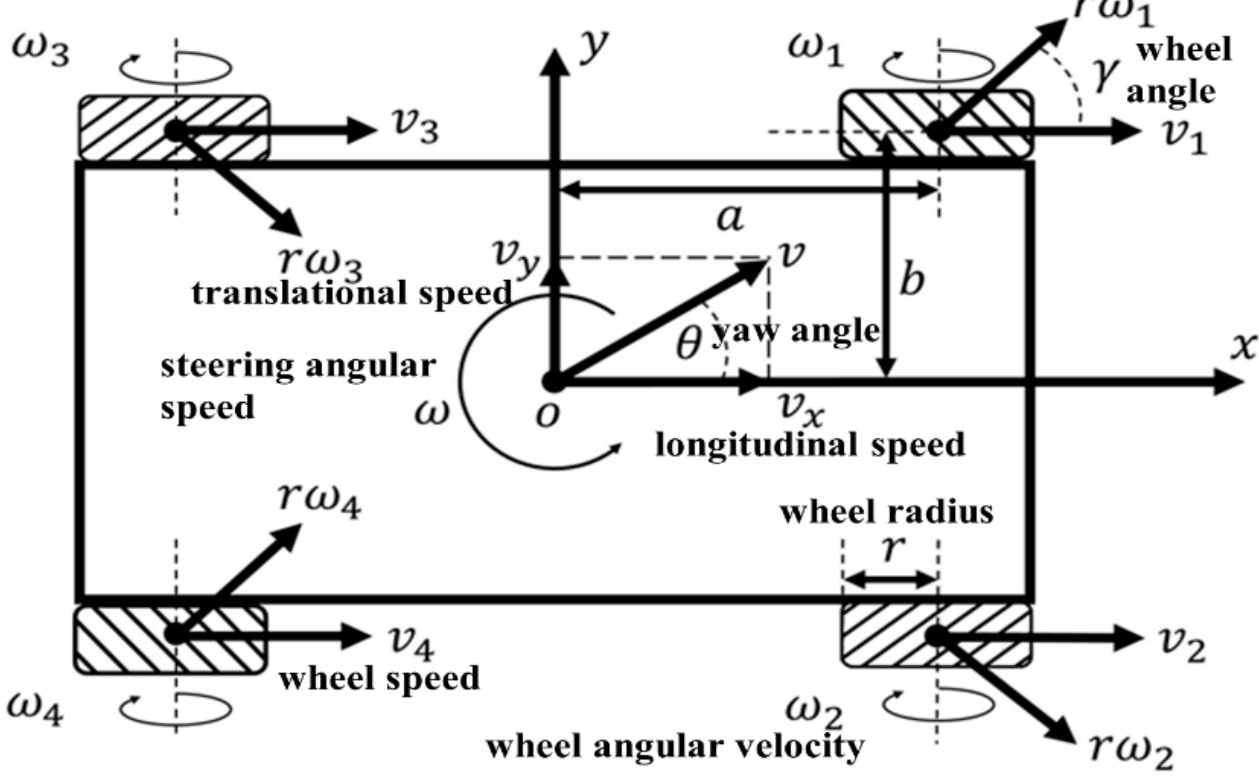

**Figure 1.** The configuration of the omnidirectional Mecanum-wheeled automated guided vehicle.

Generally, the peripheral small wheels on the omnidirectional wheel are at a constant angle of 45 degrees [7], that is, the wheels move freely at 45 degrees ($\gamma_i = ( \begin{array}{cccc} \pi/4 & -\pi/4 & -\pi/4 & \pi/4 \end{array} )$). Therefore, the motion equation of each wheel of the OMWAGV can be obtained as:

$$v_i + r\omega_i \cos(\gamma_i) = v_x - b_i\omega \tag{3}$$

$$r\omega_i \sin(\gamma_i) = v_y + a_i\omega \tag{4}$$

After rearranging Equations (3) and (4), each wheel speed $v_i$ can be calculated like this:

$$v_i = v_x - b_i\omega - \frac{v_y + a_i\omega}{\tan(\gamma_i)} \tag{5}$$

Since values of $\tan(\gamma_i), i = 1, 2, 3, 4$ in Equation (5) are 1, −1, −1, and 1, respectively, and $|a_i| = a, (i = 1, 2, 3, 4)$ and $|b_i| = b, (i = 1, 2, 3, 4)$, the speed of each omnidirectional wheel can be calculated as follows:

$$v_1 = v_x - v_y - a\omega - b\omega \tag{6}$$

$$v_2 = v_x + v_y + a\omega + b\omega \tag{7}$$

$$v_3 = v_x + v_y - a\omega - b\omega \tag{8}$$

$$v_4 = v_x - v_y + a\omega + b\omega \tag{9}$$

By rearranging Equations (6)–(9), we get:

$$v_x = r(\omega_1 + \omega_2 + \omega_3 + \omega_4)/4 \tag{10}$$

$$v_y = r(-\omega_1 + \omega_2 + \omega_3 - \omega_4)/4 \tag{11}$$

$$\omega = r(-\omega_1 + \omega_2 - \omega_3 + \omega_4)/[4(a + b)] \tag{12}$$

According to the above equations, the posture of the omnidirectional wheels can be deduced when they move. Figure 2 shows the eight main motion modes of the omnidirectional wheel vehicle, including forward, backward, left, and right translations, and four 45-degree diagonal movements.

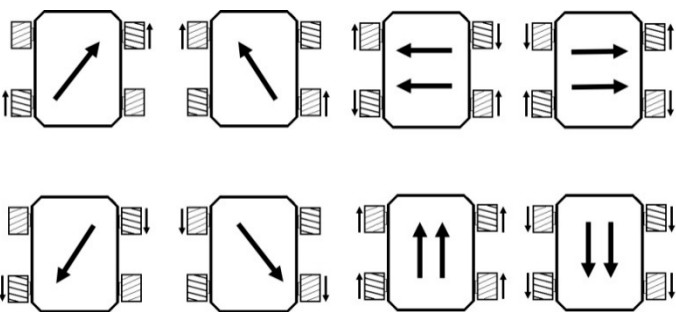

**Figure 2.** The eight main motion modes of the omnidirectional wheel vehicle.

## 3. Slip Estimation and Compensation

Assuming that the motion mode of the omnidirectional wheel is translational, $m$ is the mass of the vehicle, while $F_x$ and $F_y$ are the respective sums of the longitudinal forces and lateral forces of the four wheels. The equations for the motion of the omnidirectional wheels are:

$$m\dot{v}_x = F_x \tag{13}$$

$$m\dot{v}_y = F_y \tag{14}$$

$$F_x = (F_{d1} + F_{d2} + F_{d3} + F_{d4})/4 \tag{15}$$

$$F_y = (-F_{d1} + F_{d2} + F_{d3} - F_{d4})/4 \tag{16}$$

where $F_{di}(i = 1, 2, 3, 4)$ is the longitudinal force of each wheel. Since the speed of the OMWAGV is much slower than the speed of the general vehicle and the windward area of the OMWAGV is small, the wind resistance can be ignored. The normal force of each wheel ($F_{zi}(i = 1, 2, 3, 4)$) can be obtained as [6,12]:

$$F_{z1} = (mga - ma_x h)/2L = F_{z2} \tag{17}$$

$$F_{z3} = (mga + ma_x h)/2L = F_{z4} \tag{18}$$

where $L$ is the length from the front wheel to the rear wheel of the vehicle, $h$ is the height between the center of the vehicle and the floor, and $a_x$ is the longitudinal acceleration of the vehicle. If the vehicle is considered a quarter car model (QCM) [4], the longitudinal force of each wheel will be affected by the tire moment of inertia $I_{\omega i}$, the tire rotational angular acceleration $\dot{\omega}_i$, the tire torque $T_i$ and the tire radius $r$. The corresponding equation is as follows:

$$F_{di} = (T_i - I_{\omega i}\dot{\omega}_i)/r, \quad i = 1, 2, 3, 4 \tag{19}$$

The slip of the vehicle is the error factor between the rotation speed of the motor to the tire and the actual speed of the vehicle. This paper uses the recursive least squares (RLS) estimation method to estimate the slip value, and uses the 9-axis sensor module to capture the current speed and attitude of the vehicle. The related equation is (20) [23]:

$$v_{zn}(t) = v_n(t)\alpha_n(t) + e_n(t), \quad n = x, y \tag{20}$$

where $v_{zn}(t)$ is the speed of the vehicle measured by the 9-axis sensor, $\alpha_n(t)$ is the estimated value of slip, $v_n(t)$ is the tire speed of the vehicle, $e_n(t)$ is the measurement and estimation error, and $n = x, y$ represents the quantity in the longitudinal and lateral directions.

The error $e_n(t)$ will be:

$$e_n(t) = v_{zn}(t) - v_n(t)\alpha_n(t-1) \tag{21}$$

and slip estimation $\alpha_n(t)$ is:

$$\alpha_n(t) = \alpha_n(t-1) - K_n(t)(v_n(t)\alpha_n(t-1) - v_{zn}^*(t)) \tag{22}$$

where $v_{zn}^*(t)$ is the speed command of the vehicle. The general expression of (21) is:

$$e_n(t) = y_n(t) - \varphi_n^T(t)\Theta_n(t) \tag{23}$$

where $y_n(t)$ is the observable output of the system, $\varphi_n(t)$ is a known and measurable regressor, and $\Theta_n(t)$ is unknown but constant. $\hat{\Theta}_n(t)$ is the estimated value of $\Theta_n(t)$ at time t and meets the following least squares estimation method:

$$\min\left\{\frac{1}{2}\sum_{j=1}^{t}\eta_n^{t-j}[y_n(j) - \varphi_n^T(j-1)\hat{\Theta}_n(j)]^2\right\}, \quad 0 < \eta \le 1 \tag{24}$$

where $\eta$ is the forgetting factor, and set $dV_n(\hat{\Theta}_n(t))/d\hat{\Theta}_n(t) = 0$ is set to obtain the following results:

$$\hat{\Theta}_n(t) = P_n(t)\varphi_n(t-1)e_n(t) = \hat{\Theta}_n(t-1) + K_n(t)e_n(t)$$
$$K_n(t) = \frac{P_n(t-1)\varphi_n(t)}{\eta_n + \varphi_n^T(t)P_n(t-1)\varphi_n(t)} \tag{25}$$
$$P_n(t) = \frac{1}{\eta_n}[P_n(t-1) - \frac{P_n(t-1)\varphi_n(t)\varphi_n^T(t)P_n(t-1)}{\eta_n + \varphi_n^T(t)P_n(t-1)\varphi_n(t)}]$$

where $K_n(t)$ is the covariance matrix.

Equations (26) and (27) will estimate the slip for each wheel of the OMWAGV:

$$v_x = \frac{1}{4}\sum_{i=1}^{4}(v_{xi}\alpha_{xi} + e_{xi}) \tag{26}$$

$$v_y = [-(v_{x1}\alpha_{x1} + e_{x1}) + (v_{x2}\alpha_{x2} + e_{x2}) + (v_{x3}\alpha_{x3} + e_{x3}) - (v_{x4}\alpha_{x4} + e_{x4})]/4 \tag{27}$$

Based on the definition of the slip ratio (28) [12]:

$$\lambda = \frac{(r\omega - v)}{\max(r\omega, v)} \tag{28}$$

we may have the related slip ratio ($\lambda$) for the OMWAGV:

$$\lambda_{ni} = \frac{v_{ni}(1 - \alpha_{ni})}{\max(v_{ni}, \alpha_{ni}v_{ni})}, \quad n \in \{x, y\}, i \in \{1, 2, 3, 4\} \tag{29}$$

Different motion modes of the OMWAGV will result in different wheel speed compensation. In this paper, the slip compensation of the OMWAGV is divided into straight motion, translation, and diagonal motion. When the OMWAGV walks straight, the longitudinal speed of the vehicle is calculated by (10) and the ideal lateral speed will be zero. As the vehicle slips, there is an error between the longitudinal speed and the target speed of the vehicle, and then the vehicle will generate a lateral speed. Let $v_{zxi}^*, i = 1, 2, 3, 4$ be the longitudinal speed command of each wheel:

$$v_{zxi}^* = v_x + \lambda_{xi}v'_{xi}, \quad i = 1, 2, 3, 4 \quad v_{zn}(t) = v_n(t)\alpha_n(t) + e_n(t), \quad n = x, y \tag{30}$$

where $\lambda_{xi}$ and $v'_{xi}$ are the slip ratio and the compensation of longitudinal slip of each wheel when the OMWAGV walks straight, and:

$$v_{xi}^* = (v'_{zxi} - v_x)/\lambda_{xi}, \quad i = 1, 2, 3, 4 \tag{31}$$

When the OMWAGV walks straight, the lateral slip value of each wheel of the vehicle is 1. From Equation (11), it is known that the lateral velocity of the OMWAGV is affected by the speed of each wheel, while (32) is the compensation amount for the lateral slip of each wheel of the OMWAGV:

$$v'_{yi} = \begin{cases} (-v^*_{zyi} - v_y)(1 - \lambda_{yi}), & i = 1,4 \\ (v^*_{zyi} - v_y)(1 - \lambda_{yi}), & i = 2,3 \end{cases} \tag{32}$$

where $v^*_{zyi}$ and $\lambda_{yi}$ are the lateral speed command and slip ratio of each wheel when the OMWAGV walks straight.

When the OMWAGV translates, the lateral speed of the vehicle is calculated using (11), and the ideal longitudinal speed will be zero. When the vehicle slips, the lateral speed of the vehicle will not meet the target speed and will generate longitudinal speed. The compensation for lateral slip of the OMWAGV will be:

$$v'_{yi} = \begin{cases} (-v^*_{zyi} - v_y)/(1 - \lambda_{yi}), & i = 1,4 \\ (v^*_{zyi} - v_y)/(1 - \lambda_{yi}), & i = 2,3 \end{cases} \tag{33}$$

When the OMWAGV translates, the longitudinal slip value of each wheel of the vehicle is 1. From Equations (10) and (11), it is known that the lateral speed and longitudinal speed of the OMWAGV are affected by the speed of each wheel. The longitudinal slip compensation for the translation of the OMWAGV is:

$$v'_{xi} = \begin{cases} -(v^*_{zxi} + v_x)(1 - \lambda_{xi}), & i = 1,4 \\ (v^*_{zxi} - v_x)(1 - \lambda_{xi}), & i = 2,3 \end{cases} \tag{34}$$

The ideal speed of the OMWAGV for 45-degree oblique walking is half of the longitudinal speed and half of the lateral speed of the vehicle. The compensation for the oblique walking of the OMWAGV is:

$$v'_{xi} = \begin{cases} \frac{0.5v^*_{zxi} - v_x}{1 - 2\lambda_{xi}}, & i = 1,4 \\ \frac{0.5v^*_{zxi} - v_x}{1 - 2\lambda_{xi}}, & i = 2,3 \end{cases} \quad v'_{yi} = \begin{cases} \frac{-0.5v^*_{zyi} - v_y}{1 - 2\lambda_{yi}}, & i = 1,4 \\ \frac{0.5v^*_{zyi} - v_y}{1 - 2\lambda_{yi}}, & i = 2,3 \end{cases} \tag{35}$$

## 4. Experimental Results

The proposed system includes four sets of brushless DC motors, gearboxes, controllers, omnidirectional Mecanum wheels, and sensors. The brushless DC motors are 5RB100KS-SY10 with encoders that calculate the actual speed of the motor ($\omega_i (i = 1, 2, 3, 4)$). The Hall sensor signals are fed back to the PID speed controller for motor position and speed control. A gearbox with a 19:1 gear ratio is used to increase the motor torque, so the maximum motor speed is 157.89 r/min and the rated torque is 7.6 Nm. An MPU-9265 [24] is installed on the vehicle to measure the actual speed and direction of the vehicle ($v_x, v_y, \theta$), estimate the slip of the OMWAGV, and make the robot move more accurately. The MPU-9250 is a multi-chip module nine-axis motion tracking device, which consists of a three-axis gyroscope, a three-axis accelerometer, a three-axis magnetometer, and a digital motion processor (DMP). In addition, the MPU-9250 also provides nine 16-bit analog-to-digital converters (ADCs) for digitizing nine-axis analog outputs. With the inputs from the encoders and the MPU-9250, a microprocessor (dsPIC30F6010A) [25] is used to program software that estimates the slip by the RLS method, calculate the slip ratio ($\lambda_x, \lambda_y$), and control the motion of the OMWAGV. The slip compensation block with the input of slip ratio will calculate the compensation value, which is used for the slip error compensation control of the OMWAGV. The block diagram of slip error compensation control for the OMWAGV is shown in Figure 3. Figure 4 shows the appearance of the OMWAGV. Figure 5 is the hardware of the experimental system. The size of the vehicle is 90 cm long, 50 cm wide, 54 cm high, and the weight is about 60 kg.

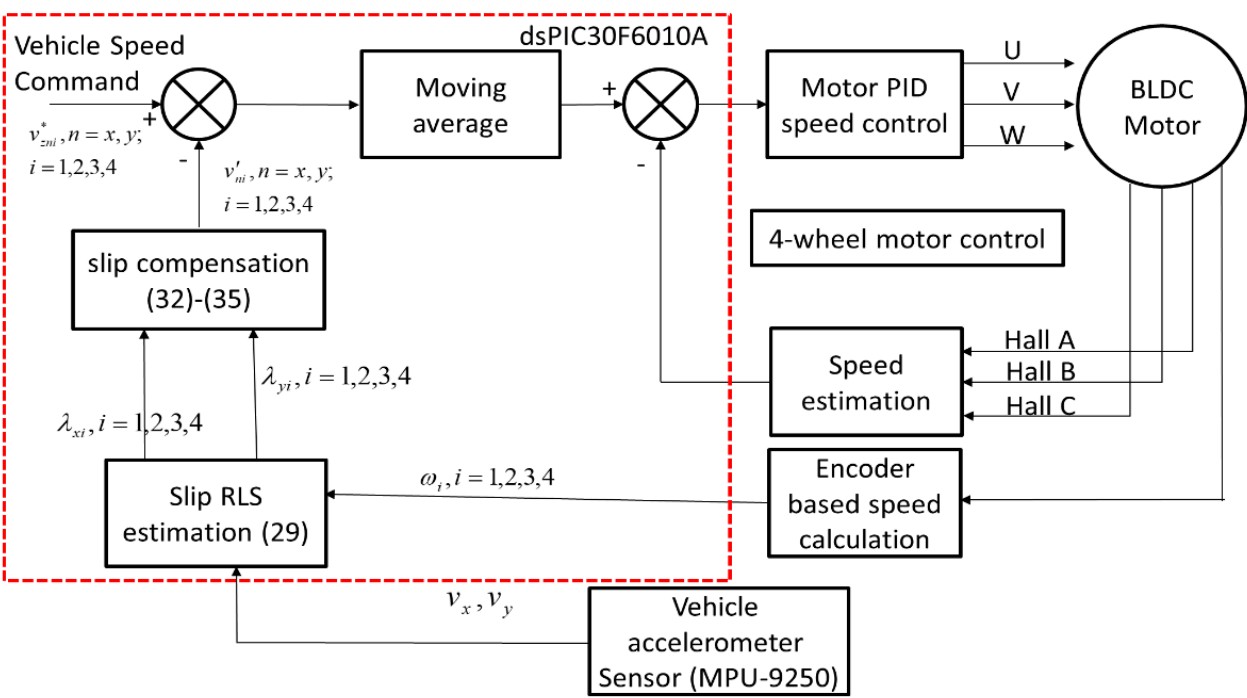

**Figure 3.** The block diagram of slip error compensation control for the omnidirectional Mecanum-wheeled automated guided vehicle driven by BLDC motors.

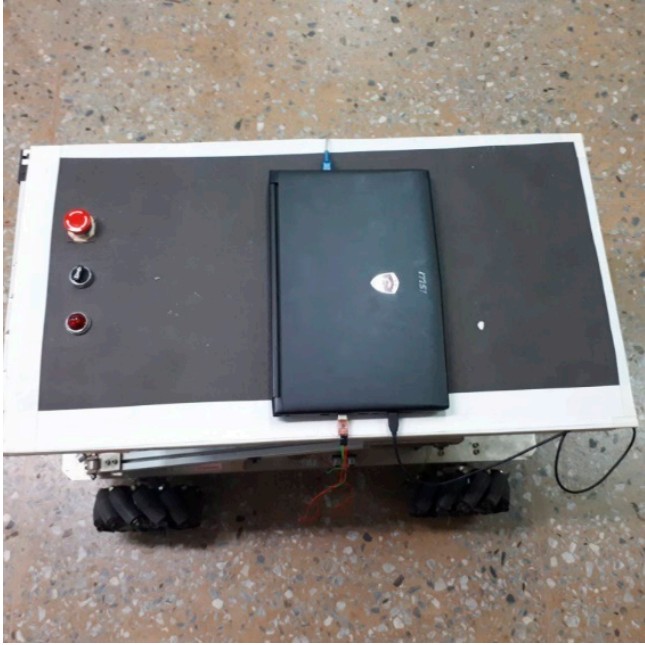

**Figure 4.** The appearance of the omnidirectional Mecanum-wheeled automated guided vehicle.

As the vehicle speed is set to 30 cm/s, Figures 6–9 show the four motor speeds, the vehicle speed, and the estimated values of the RLS method. The blue and yellow curves in the lower part of the figures respectively stand for the speed of the vehicle captured by the nine-axis sensor and the wheel speeds, and the upper part is the estimated values by the RLS method using Equation (20). The means and standard deviations of the speeds and the estimated slip values in Figures 6–9 are presented in Tables 1 and 2. It is easier to compare the performance of each motor. Figure 10 shows the estimated values by the RLS method of four motors together. The performance of each motor is slightly different.

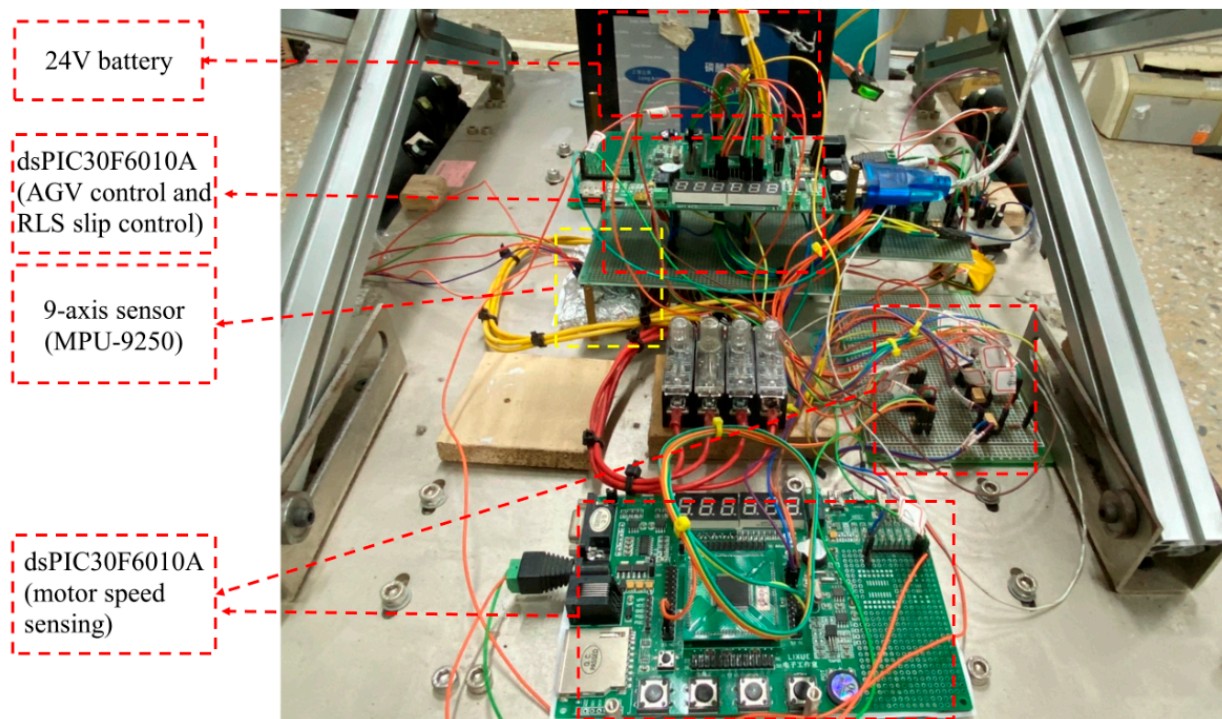

**Figure 5.** The hardware of the experimental system.

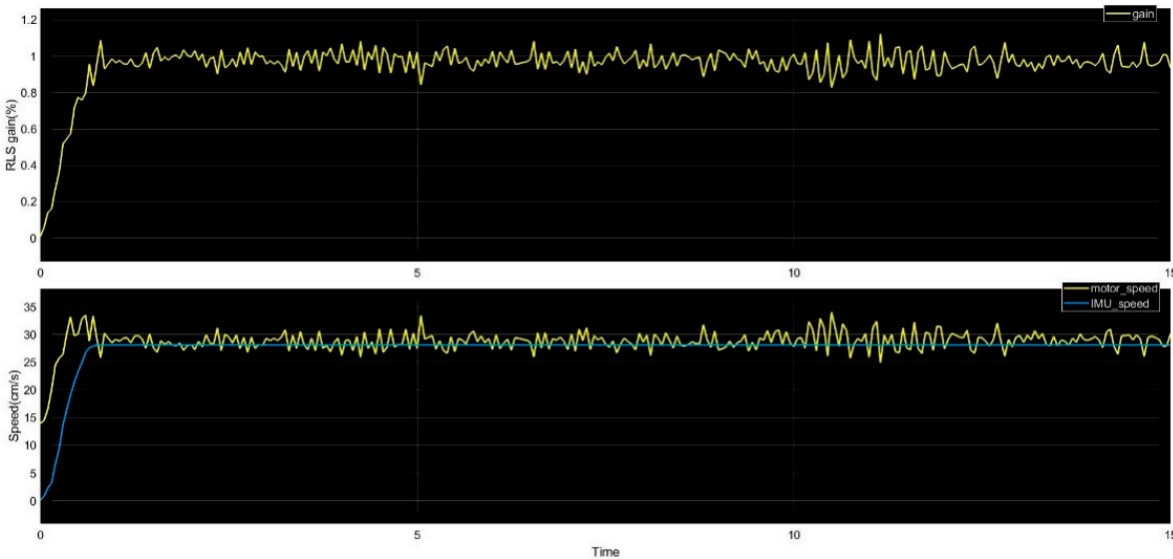

**Figure 6.** Estimated value by RLS method, motor speeds, and the vehicle speed of the first motor.

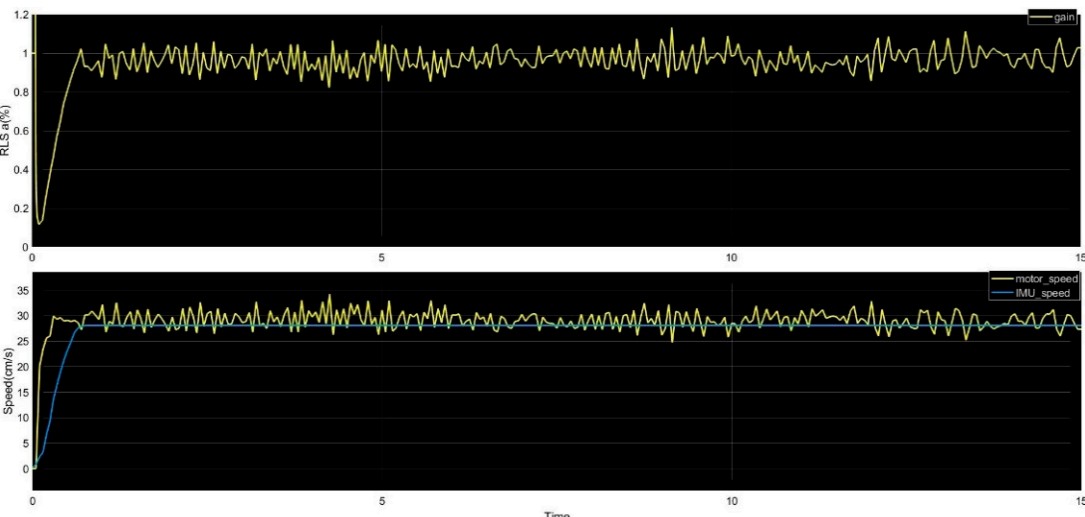

**Figure 7.** Estimated value by RLS method, motor speeds, and the vehicle speed of the second motor.

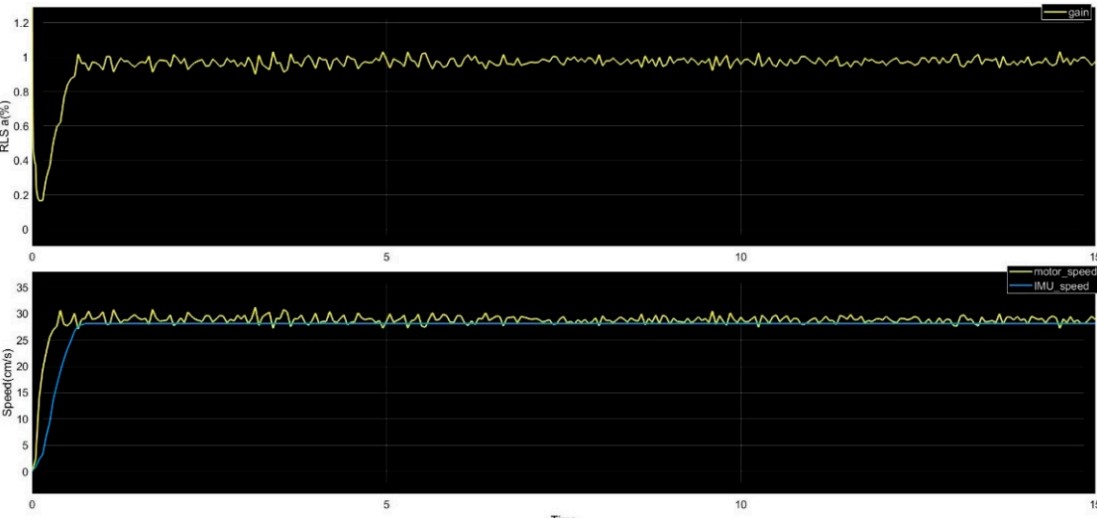

**Figure 8.** Estimated value by RLS method, motor speeds, and the vehicle speed of the third motor.

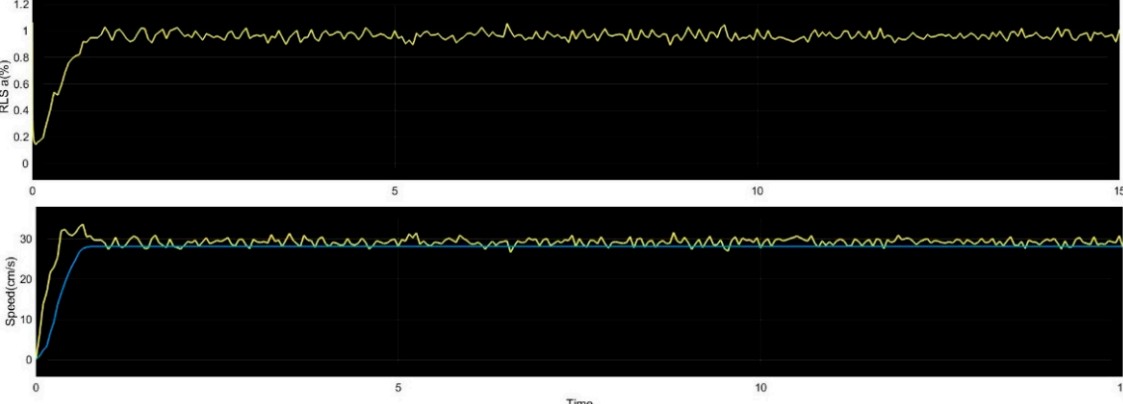

**Figure 9.** Estimated value by RLS method, motor speeds, and the vehicle speed of the fourth motor.

**Table 1.** The means and standard deviations of the speeds in Figures 6–9 (unit: cm/s).

| Speed | MPU-9250 | First Motor | Second Motor | Third Motor | Fourth Motor |
|---|---|---|---|---|---|
| mean | 28.103 | 28.633 | 29.457 | 28.971 | 29.179 |
| standard deviation | 0.023 | 1.322 | 1.960 | 0.848 | 0.945 |

**Table 2.** The means and standard deviations of the estimated slip values in Figures 6–9.

| Estimated Slip | First Motor | Second Motor | Third Motor | Fourth Motor |
|---|---|---|---|---|
| mean | 0.951 | 1.036 | 0.945 | 0.910 |
| standard deviation | 0.001 | 0.002 | 0.002 | 0.002 |

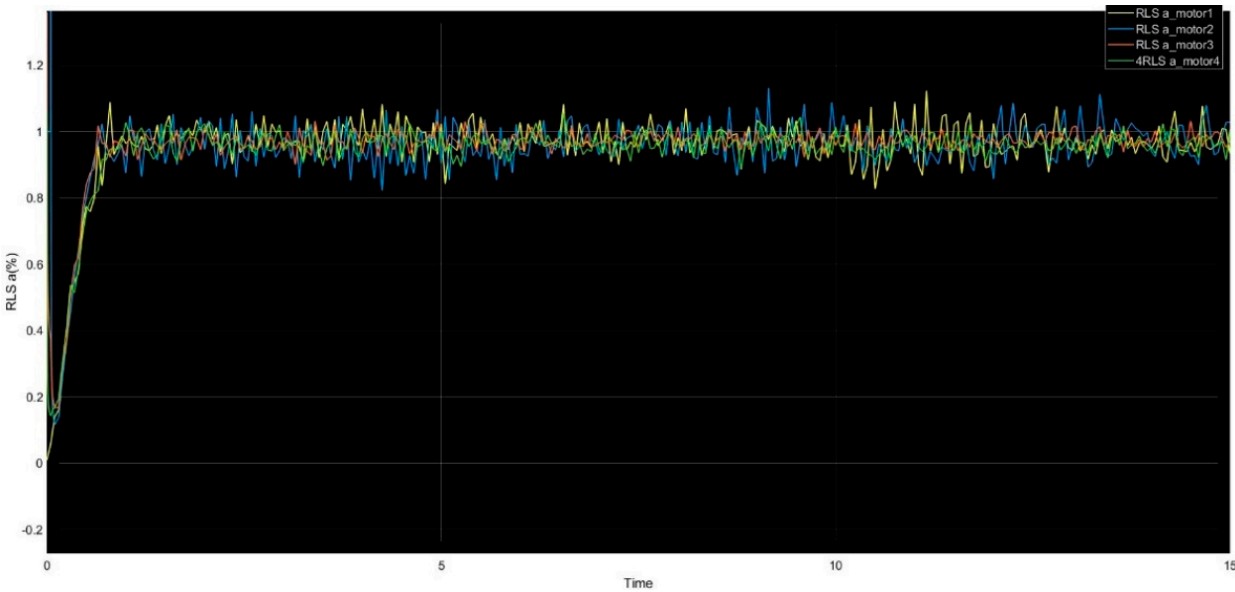

**Figure 10.** Estimated values by RLS method of four motors.

Figure 11 shows the actual slip ratio of the fourth motor (top), the estimated slip ratio (middle), and their difference (bottom). The actual slip ratio is the $\lambda$ value from Equation (28), and the estimated value is obtained by substituting the estimated value of the RLS method (22) into Equation (29). The other three motors have similar results.

Figure 12 shows the test site for walking straight, where the green tape stands for the starting point, the yellow one is 50 cm, and the red ones are 100 cm, 200 cm, 300 cm, and 400 cm, respectively. The walking distances of 100 cm and 400 cm are conducted. The steps of the OMWAGV moving 100 cm without and with using RLS estimation compensation are shown in Figures 13 and 14. It can be seen that the OMWAGV has little error when walking short distances. Figures 15 and 16 show the case of walking 400 cm without and with RLS estimation compensation, respectively. In Figure 15, it is obvious that the OMWAGV is slowly shifting to the right and towards the end during walking. The OMWAGV deviated from the white line by 16 cm and also exceeded the end. In Figure 16, after walking, the robot deviated from the white line by 4 cm and the error at the end was less than 1 cm.

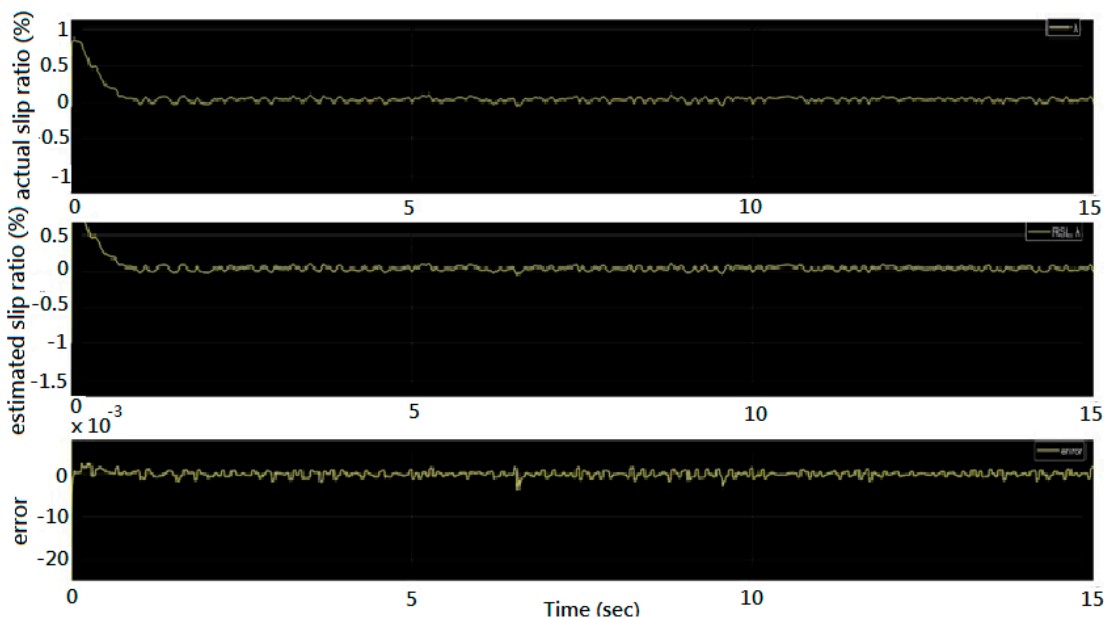

**Figure 11.** The actual slip ratio (**top**), the estimated slip ratio (**middle**), and difference (**bottom**) of the fourth motor.

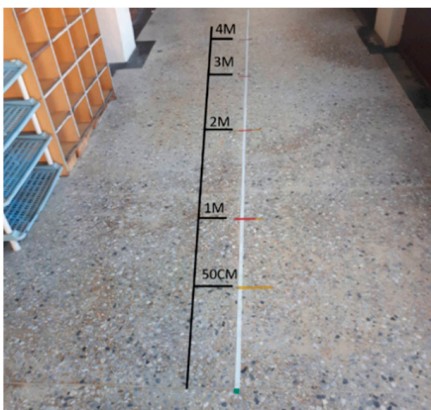

**Figure 12.** The test site for walking straight.

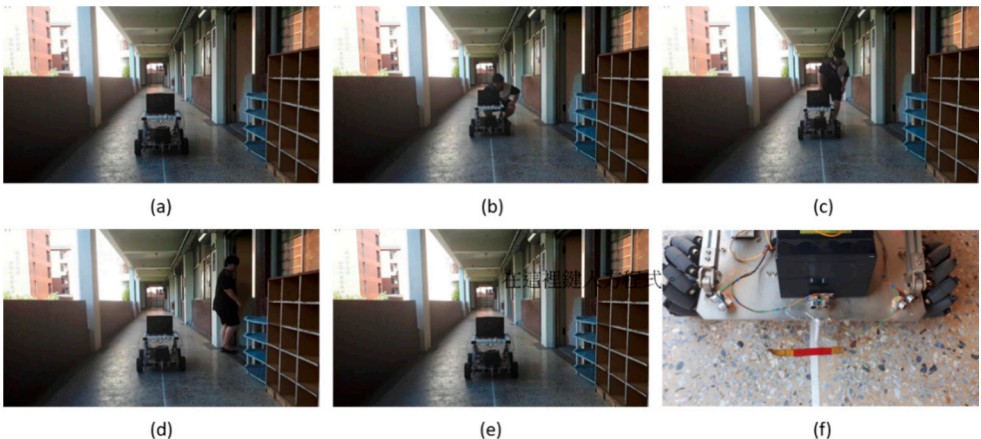

**Figure 13.** Omnidirectional Mecanum-wheeled automated guided vehicle (OMWAGV) walking straight 100 cm without RLS estimation compensation. (**a**) Starting position, (**b**–**e**) at middle way, and (**f**) stop position.

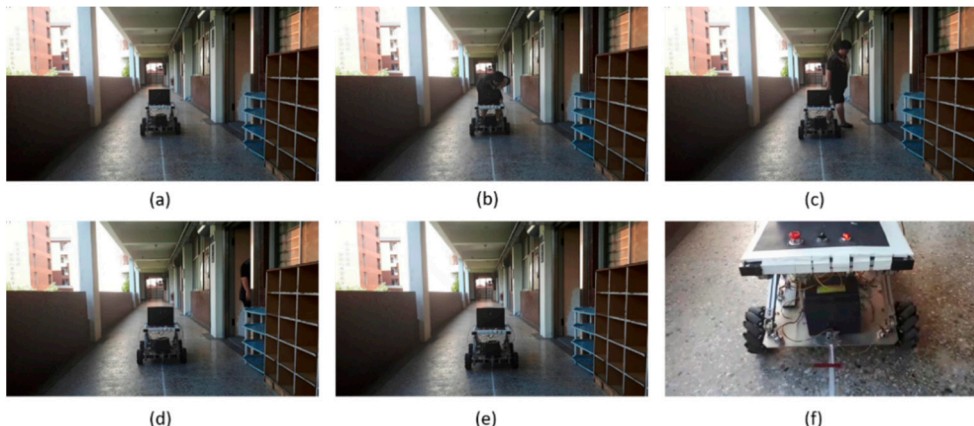

**Figure 14.** OMWAGV walking straight 100 cm with RLS estimation compensation. (**a**) Starting position, (**b**–**e**) at middle way, and (**f**) stop position.

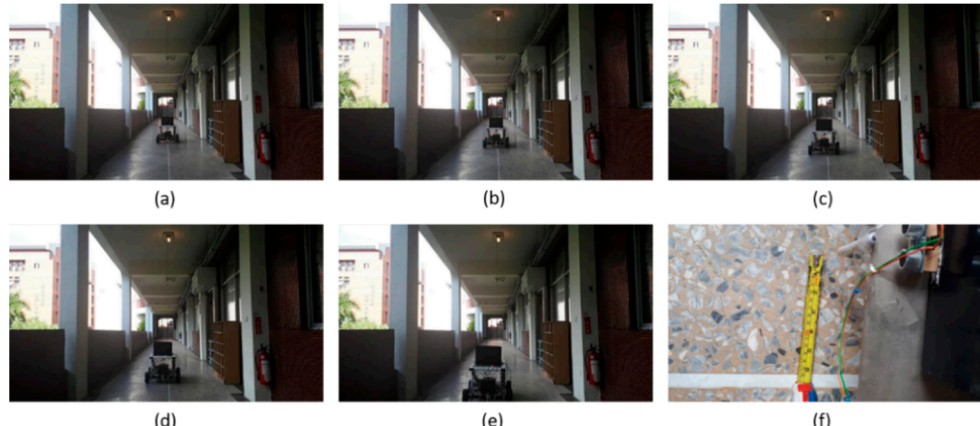

**Figure 15.** OMWAGV walking straight 400 cm without RLS estimation compensation. (**a**) Starting position, (**b**–**e**) at middle way, and (**f**) deviation at stop position.

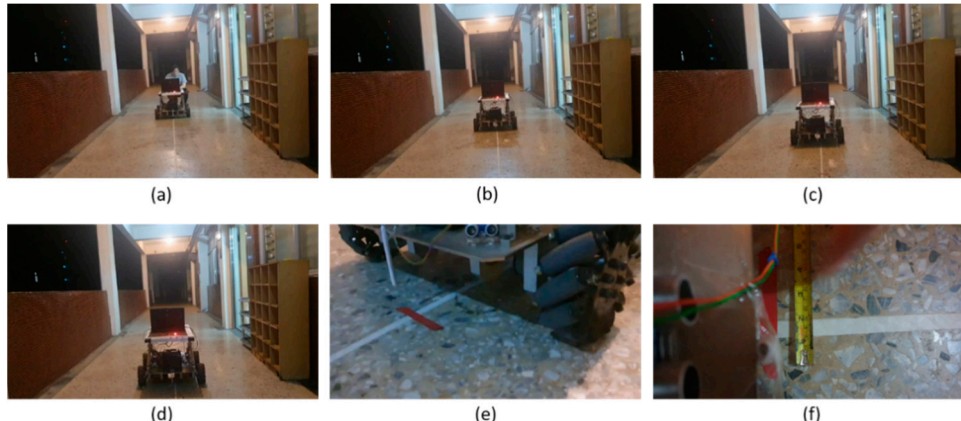

**Figure 16.** OMWAGV walking straight 400 cm with RLS estimation compensation. (**a**) Starting position, (**b**–**e**) at middle way, and (**f**) deviation at stop position.

Figure 17 shows the captured pictures of the OMWAGV moving diagonally without using RLS estimation compensation. The process is from the starting position, forward at an angle of 45 degrees to the left, back to the starting position, backward at an angle of 45 degrees to the right, back to the starting position, forward at an angle of 45 degrees to the right, back to the starting position, backward at an angle of 45 degrees to the left,

and finally back to the starting position. Figure 18 displays the path by software package Tracker [26], and clearly shows that the path of the vehicle has shifted significantly, where the yellow point is the starting position and the black point is the end position. Tracker is a free video analysis and modeling tool built on the Open Source Physics (OSP) Java framework. It is designed to be used in physics education. Tracker video modeling is a powerful way to combine videos with computer modeling. Figure 19 displays the captured pictures of the OMWAGV with RLS estimation and compensation for moving diagonally. Figure 20 shows the walking path of the video of Figure 19. Comparing the paths of Figures 18 and 20, the deviation of the latter is greatly reduced. Since the shooting is not directly above the scene, the displayed walking paths are distorted. Figures 21 and 22 are the captured pictures of the robot without and with RLS estimation and compensation for cross-walking motion. Figures 23 and 24 show the walking paths of Figures 21 and 22 by Tracker, and the travel deviation of the latter is greatly reduced. In summary, the distances between the original starting position to the stopping position are 1.52 m, 0.03 m, 1.56 m, and 0.03 m in Figures 18, 20, 22 and 24, respectively. The effectiveness of the proposed compensation and control algorithm are clear.

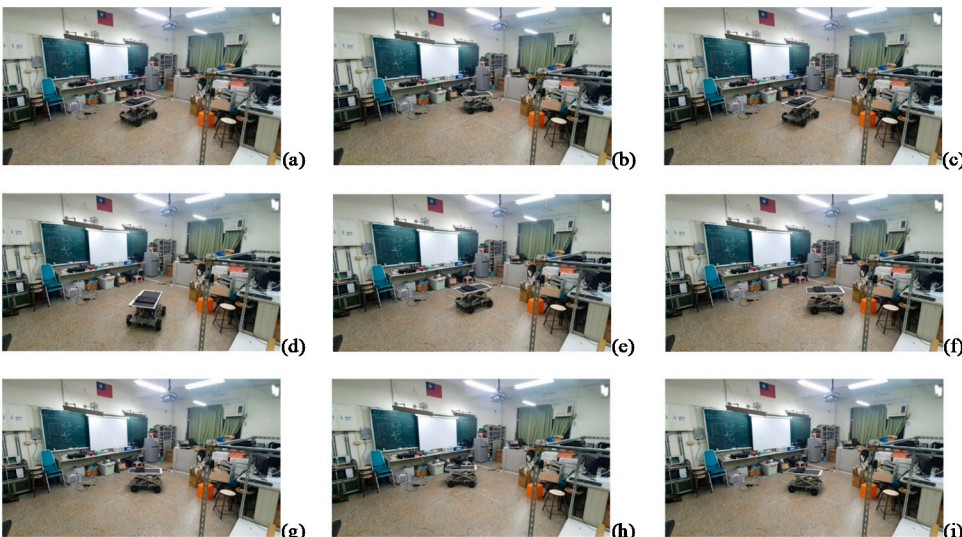

**Figure 17.** The captured pictures of the OMWAGV moving diagonally without RLS estimation and compensation. (**a**) Starting position, (**b**) forward at an angle of 45 degrees to the left, (**c**) return to the starting position, (**d**) backward at an angle of 45 degrees to the right, (**e**) back to the starting position, (**f**) forward at an angle of 45 degrees to the right, (**g**) return to the starting position, (**h**) backward at an angle of 45 degrees to the left, (**i**) return to the starting position.

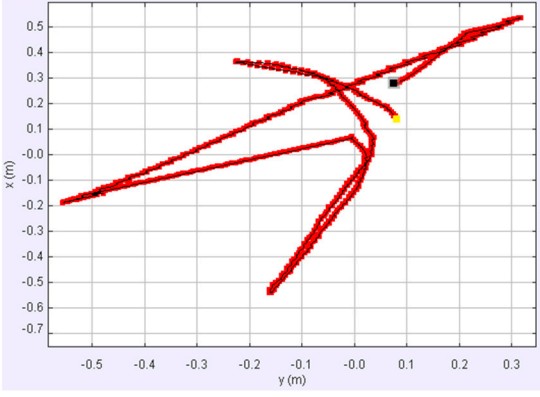

**Figure 18.** The path in Figure 17 shown by Tracker.

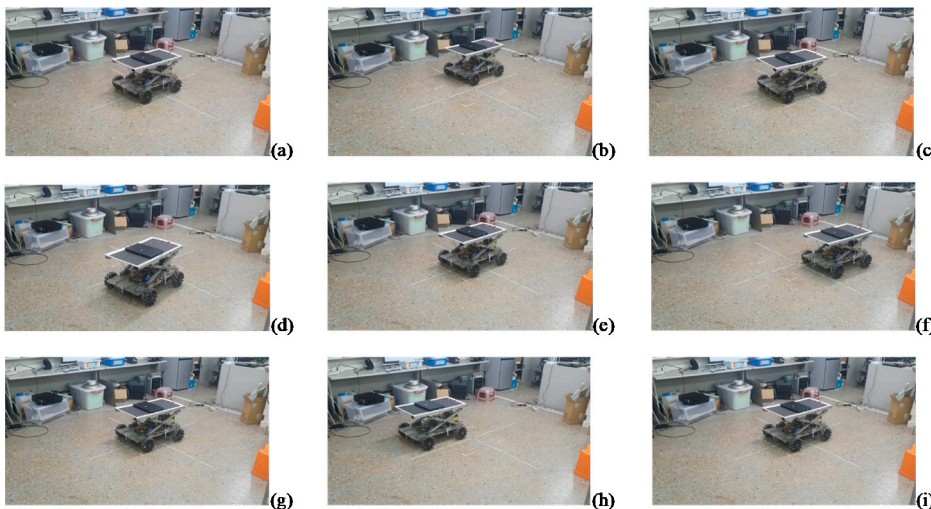

**Figure 19.** The captured pictures of the OMWAGV moving diagonally with RLS estimation compensation. (**a**) Starting position, (**b**) forward at an angle of 45 degrees to the left, (**c**) return to the starting position, (**d**) backward at an angle of 45 degrees to the right, (**e**) back to the starting position, (**f**) forward at an angle of 45 degrees to the right, (**g**) return to the starting position, (**h**) backward at an angle of 45 degrees to the left, (**i**) return to the starting position.

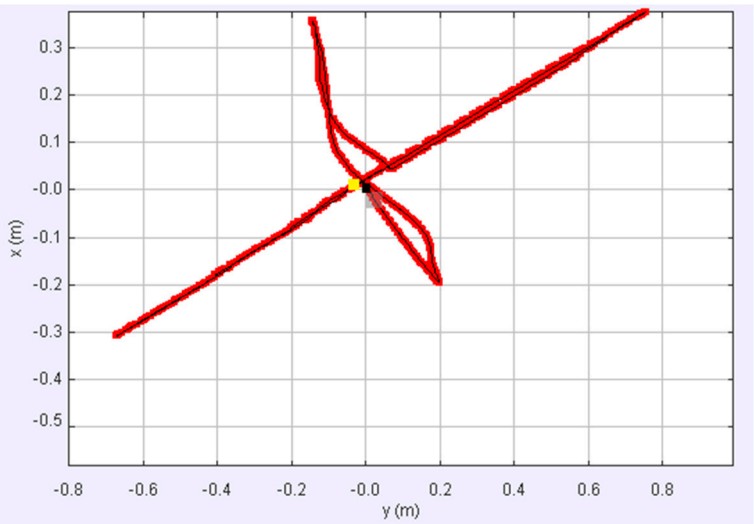

**Figure 20.** The path in Figure 19 shown by Tracker.

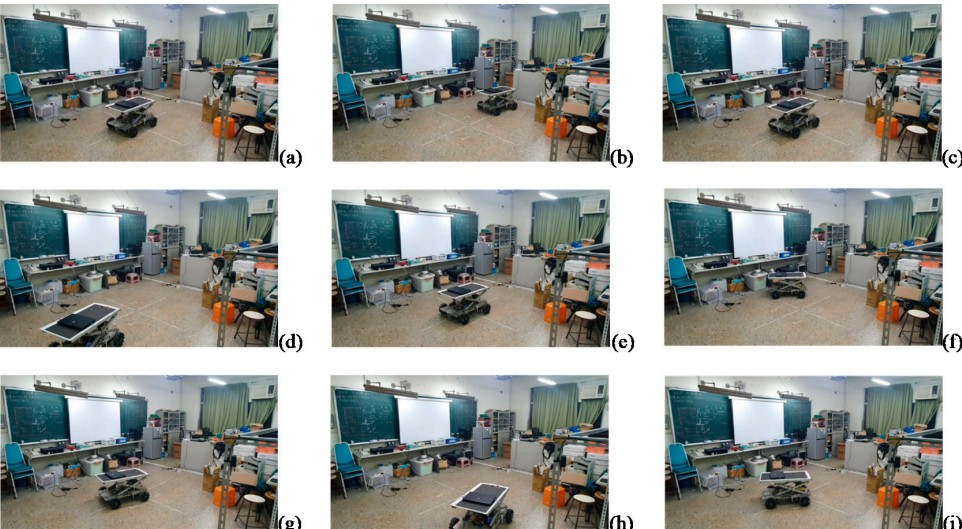

**Figure 21.** The captured pictures of the robot's cross-walking motion without RLS estimation and compensation. (**a**) Starting position, (**b**) forward, (**c**) back to the starting position, (**d**) backward, (**e**) back to the starting position, (**f**) left shift, (**g**) back to the starting position, (**h**) right shift, (**i**) back to the starting position.

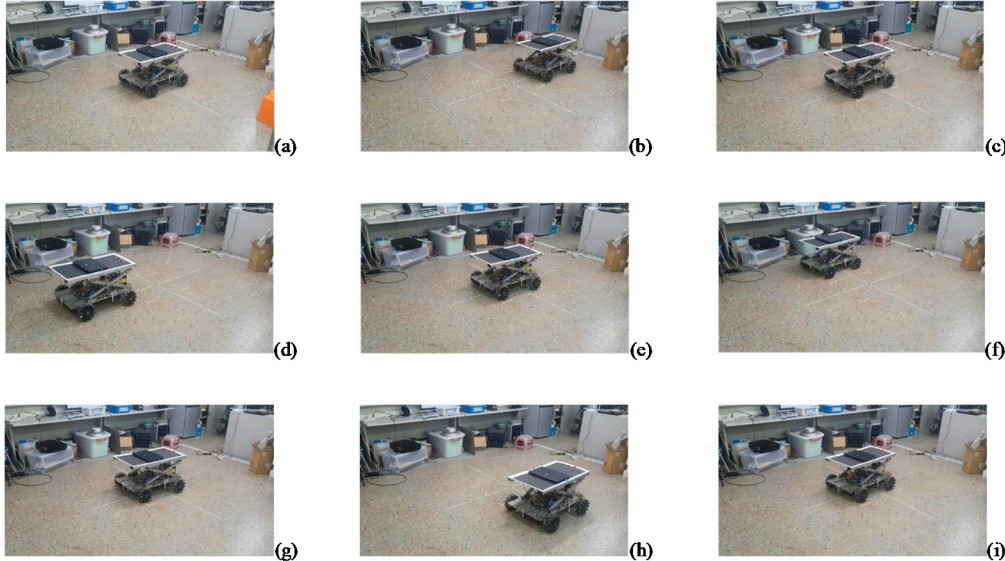

**Figure 22.** The captured pictures of the robot's cross-walking motion with RLS estimation and compensation. (**a**) Starting position, (**b**) forward, (**c**) back to the starting position, (**d**) backward, (**e**) back to the starting position, (**f**) left shift, (**g**) back to the starting position, (**h**) right shift, (**i**) back to the starting position.

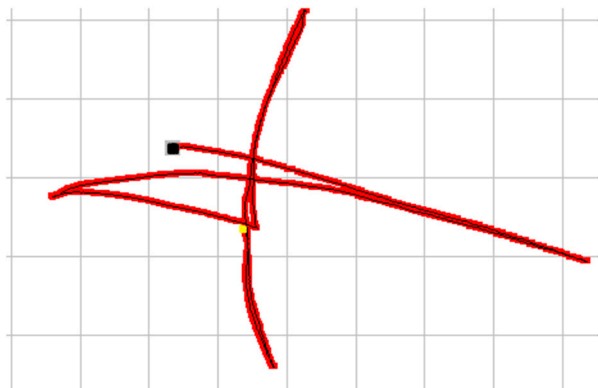

**Figure 23.** The path in Figure 21 shown by Tracker.

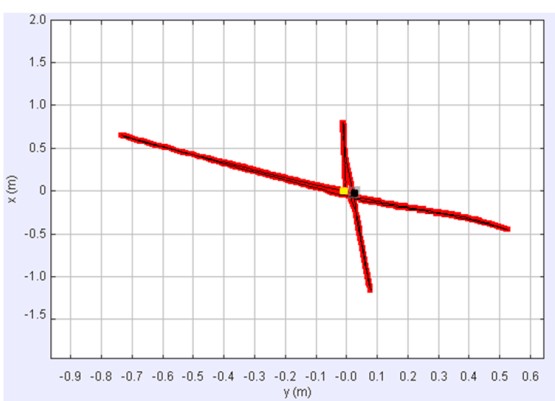

**Figure 24.** The path in Figure 23 shown by Tracker.

Figures 25 and 26 show the OMWAGV with a load of 46 kg walking a distance of 400 cm without and with RLS estimation and compensation, respectively. In the former, it is obvious that the robot was slowly shifting to the right during walking and deviated from the white line by 38 cm. In the latter, the robot deviated from the white line by 4 cm and the error at the end was less than 1 cm.

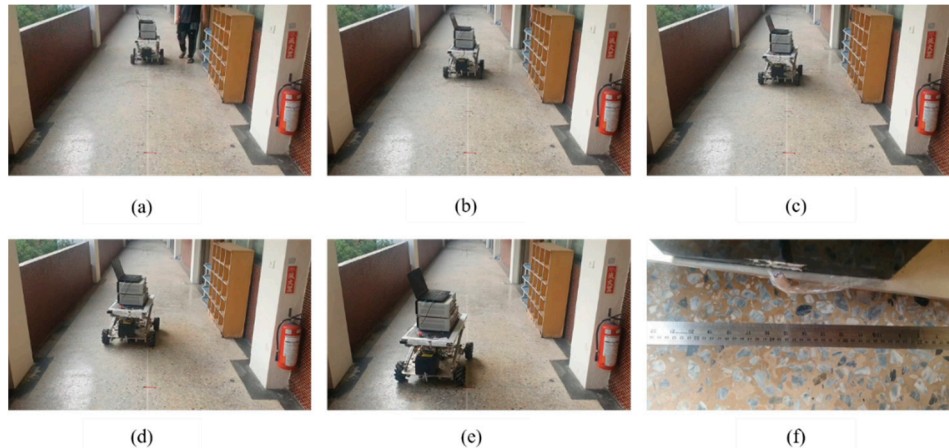

**Figure 25.** The OMWAGV with a load of 46 kg walking a distance of 400 cm without RLS estimation and compensation. (**a**) Starting position, (**b**–**e**) at middle way, and (**f**) deviation at stop position.

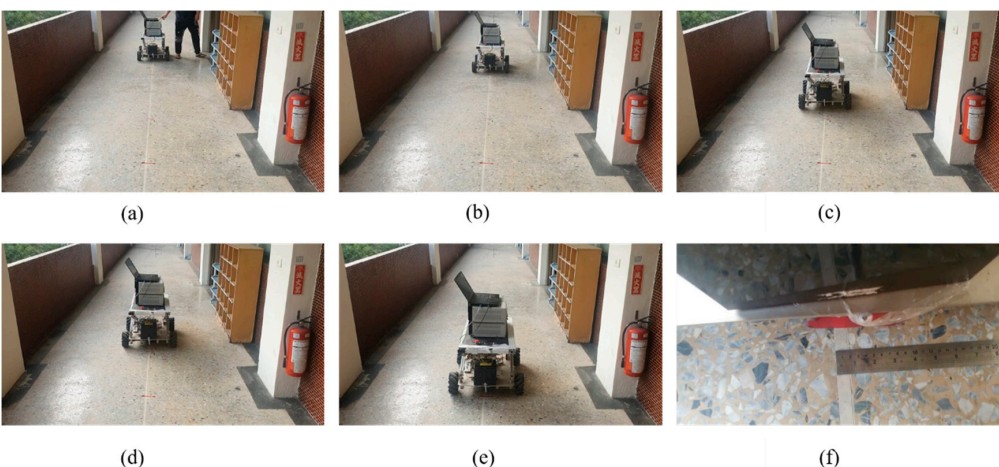

**Figure 26.** The OMWAGV with a load of 46 kg walking a distance of 400 cm with RLS estimation and compensation. (**a**) Starting position, (**b**–**e**) at middle way, and (**f**) deviation at stop position.

## 5. Conclusions

In this paper, an OMWAGV with slip estimation and compensation control is designed and implemented. The model of the OMWAGV is first introduced. Based on a quarter car model for the OMWAGV, a recursive least squares (RLS) method is described and utilized to estimate real-time slip ratio changes for four driving wheels. The slip ratio compensation and control of the OMWAGV is considered for straight motion, translation, and diagonal motion. The proposed system includes a Microchip dsPIC30F6010A, a nine-axis accelerometer sensor (MPU-9250), four Mecanum wheels, and four sets of brushless DC motors, gearboxes, and controllers. The experimental results of diagonally moving and cross-walking motion without and with slip estimation and compensation control show that, without calculating the errors that occur during travel, the distances between the original starting position and the stopping position are dramatically reduced from 1.52 m to 0.03 m and from 1.56 m to 0.03 m, respectively. The higher tracking accuracy of the proposed method has verified its effectiveness and validness. Referring to [27], the performance comparison between the proposed method and other novel control methods is indispensable for us in the near future.

**Author Contributions:** P.-J.C., Y.-P.C., M.M. and M.-S.W. conceived and designed the experiments; Y.-P.C., M.M., and S.-Y.Y. performed the experiments; M.-S.W. and S.-Y.Y. analyzed the data; M.-S.W. and P.-J.C. contributed materials and analytical tools; M.-S.W. wrote the paper. All authors have read and agreed to the published version of the manuscript.

**Funding:** This research was funded by the Higher Education Sprout program of the Ministry of Education, Taiwan.

**Conflicts of Interest:** The authors declare no conflict of interest. The funders had no role in the design of the study, the collection, analyses, or interpretation of data, the writing of the manuscript, or the decision to publish the results.

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
