# Peer review of "Slip Estimation and Compensation Control of Omnidirectional Wheeled Automated Guided Vehicle"

_electronics, doi:10.3390/electronics10070840_

Round 1

Reviewer 1 Report

  • Good research very well written.
  • Great job periodically repeating the definition of the acronym for omnidirectional wheeled automated guided vehicle. Doing so will make reading easier.  Please consider adding it to section 2’ s title, so the reader instantly understands the content of the section, and please consider doing so for figure 3.
  • Figure 1 is very good and transmits considerable information to the reader. Please consider adding variable definitions to the figure caption to increase readability.
  • Please elaborate what should be sought respectively in the doubly-cited [15,16], or in the event the citation is duplicative, please limit the readers’ labors by choosing the single best citation.
  • MAJOR WEAKNESS: Results are qualitative only. Please present quantitative results in a table (or several). The reviewer recommends inclusion for each case iterated, the estimation accuracies (means and standard deviations of the results plotted in figures 6-11), path tracking accuracy (means and standard deviations of results plotted in figures 18, 20, 22, and 24).  
  • The conclusions are very short and include no broadly termed quantitative results. Using the newly added tables of data mentioned above, please declare a case the “baseline”, and then mention percent improvement from the baseline case, adding this verbiage to the conclusions and also the abstract (representing “results in broadest possible terms” as described in the manuscript template).
  • Another common enhancement to Conclusions sections is to (very briefly) describe the authors’ thoughts on future research. The reviewer recommends comparison of the proposed methods with the novel motor control methods just published in Applied Sciences, Sands, T. Control of DC Motors to Guide Unmanned Underwater Vehicles. Sci. 2021, 11(5), 2144. https://doi.org/10.3390/app11052144

Author Response

Please find our responses as attached.

Reviewer 2 Report

The content of the article is consistent with the scientific area of the journal Electronics. The subject raised by the authors is current and so far rarely noticed by other authors publishing in this area. The issue described may in the future contribute to improving the efficiency of the automation and of the omni directional mecanum-wheeled automated guided vehicle or recursive 35 least square and slip ratio.
The paper has an original, scientific character, compensation control of omnidirectional wheeled automated guided vehicle. This paper proposed a slip estimation and compensation control method for an omnidirectional Mecanum-wheeled automated guided vehicle (OWAGV) and its implementation of software and hardware control system. The proposed system uses microcontroller and a 9-axis accelerometer sensor.
For a better clarification, please edit your paper as follows: 1. Extend the text of manuscript (example introduction or conclusion) to concrete results in the world and in Europe, - Improve the quality of the paper by presenting the results of publications of researchers and experts that are registered in the world databases (wos). They are specifically these: Experimental investigations of a highly maneuverable mobile omniwheel robot, Integration of Inertial Sensor Data into Control of the Mobile Platform and Navigation control and stability investigation of a mobile robot based on a hexacopter equipped with an integrated manipulator. Thanks. 2. figure 3 should be contrasting and readable, 3. conclusions and future work should be extended to contain practical applications based on research described in this paper - expand references, 4. modify the mathematical expression (formula) No: 5, 
5. the paper should be read by a native english speaker. 
After consideration of these minor comments, the article is properly prepared (in the reviewer opinion) for publication in the journal  Electronics

Author Response

Please find our responses as attached.

Reviewer 3 Report

The contribution of this work is interesting and deserves publication. The title is descriptive. The abstract clearly indicates the scope. The paper is well organised and logically written, nevertheless, the English language of the contribution should be improved by a native speaker.

Appropriate research goals are chosen in this contribution, which shows that the authors have a high level of understanding of current research within the field. I suggest that the authors explain more in depth the choice of the. The presentation of the results in terms of the research objectives has been  made, nevertheless, there should be a deeper clarification.

The authors have been able to draw logical conclusions from the results.

The quality of pictures and figures is good.

Nevertheless the authors should address the problem of the stability of the whole control scheme. Moreover, please address also the convergence of the slip estimation using the RLS procedure. Is the estimation procedures always convergent?

Pease clarify these points.

Concerning the cited literature, please consider also the following papers which can help to orient the readers in the context of this topic.

Mercorelli, P. Parameters identification in a permanent magnet three-phase synchronous motor of a city-bus for an intelligent drive assistant (2014) International Journal of Modelling, Identification and Control, 21 (4), pp. 352-361. Schimmack, M. et al. Scaling-based least squares methods with implemented Kalman filter approach for nano-parameters identification (2016) International Journal of Modelling, Identification and Control, 25 (2), pp. 85-92. Straßberger, D. et al. A geometric approach to decouple robotino motions and its functional controllability (2015) Journal of Physics: Conference Series, 659 (1), art. no. 012027

Author Response

Please find our responses as attached.
